# Entropic Dynamics of Exchange Rates and Options

**DOI:** 10.3390/e21060586

**Published:** 2019-06-13

**Authors:** Mohammad Abedi, Daniel Bartolomeo

**Affiliations:** Department of Physics, University at Albany-SUNY, Albany, NY 12222, USA

**Keywords:** entropic inference, maximum entropy, entropic dynamics, Fokker–Planck equation, Garman–Kohlhagen model, Black–Scholes–Merton partial differential equation

## Abstract

An Entropic Dynamics of exchange rates is laid down to model the dynamics of foreign exchange rates, FX, and European Options on FX. The main objective is to represent an alternative framework to model dynamics. Entropic inference is an inductive inference framework equipped with proper tools to handle situations where incomplete information is available. Entropic Dynamics is an application of entropic inference, which is equipped with the entropic notion of time to model dynamics. The scale invariance is a symmetry of the dynamics of exchange rates, which is manifested in our formalism. To make the formalism manifestly invariant under this symmetry, we arrive at choosing the logarithm of the exchange rate as the proper variable to model. By taking into account the relevant information about the exchange rates, we derive the Geometric Brownian Motion, GBM, of the exchange rate, which is manifestly invariant under the scale transformation. Securities should be valued such that there is no arbitrage opportunity. To this end, we derive a risk-neutral measure to value European Options on FX. The resulting model is the celebrated Garman–Kohlhagen model.

## 1. Introduction

To understand, describe, and predict phenomena, scientists have come up with the scientific method of reasoning. For a small fraction of the situations where complete information about the subject is accessible logic is proposed as a framework to reason. In most situations, where they are faced with not having enough information about the system, an extension of the logic is required. To deal with such situations, an *inductive inference framework* is designed. *Entropic inference* is an inductive inference framework designed with proper tools to cope with situations where incomplete information is at our disposal [1,2,3].

The first tool of entropic inference designed to represent a quantitative description of the state of partial belief is probability theory. The probability distribution represents the information we have about the outcome of the system. The second tool is designed to handle the situation when new information about the subject matter is accessible. The relative entropy is designed such that it can incorporate the new information and *update* the state of partial knowledge, the probability distribution [4,5]. An important criterion in designing the relative entropy is the Principle of Minimal Updating. This principle ensures that the prior probability distribution is updated only to the extent required by the new information. By maximizing the relative entropy, a posterior distribution is derived, which is the least biased distribution, namely it describes all given information, nothing else. It is noteworthy to mention that this notion of entropy is not derived from physics; it is shown that the known notion of entropy in physics is an application of the relative entropy [6,7]. The third tool of inference is the information geometry. The space of probability distributions defines an information geometry with a unique metric, which defines the *distance* between two neighboring distributions [8,9]. The significance of information geometry in finance will be addressed in future work where we will model the dynamics of many equities and address how to invest [10].

A long-standing aspiration of scientists has been describing how a quantity of interest changes, the *dynamics*. The conception of time is contrived to further simplify describing any dynamics. If entropic inference is any good, as a formalism divorced from science, not derived from it, it ought to come up with a notion of time that leads to the known notions of time, i.e., the notions of time should *emerge* from the entropic framework. *Entropic Dynamics* is an application of the entropic inference framework, which defines *entropic time* and in turn enables the formalism to model dynamics. This entropic time can be tailored to suit modeling different unrelated dynamics where the entropic time pertains to the system of interest rather than having a unique universal notion of time for all branches of science [11]. Entropic Dynamics has extensively been applied to model dynamics in physics [12,13,14,15,16,17,18].

Modeling the dynamics of securities, especially stocks, goes back to Bachelier’s thesis [19]. For quite a while, his brilliant work was forgotten until it was rediscovered by Samuelson [20]. Models were developed in the regime of continuous dynamics where no jump happens [19,20,21]. The resulting Geometric Brownian Motion dynamics was used by Black and Scholes and later by Merton to value Options [22,23,24]. The models were extended in many directions such as including stochastic volatility [25,26,27,28,29,30,31,32,33,34]. Pricing European Options on the exchange rate was developed by Garman and Kohlhagen [35]. In later works, the Garman–Kohlhagen model was extended in various directions [36,37,38,39].

We present an *alternative framework* to model dynamics. In our formalism, the dynamical models are derived by maximizing the relative entropy. The relative entropy is *designed* as a tool of inference to update the state of partial belief when new information is available. We do not resort to a principle such as an action principle or Hamiltonian mechanism, which is useful in a particular branch of science, to derive dynamics, nor do we assume the dynamics like in the stochastic mechanics formalism. The significance of our formalism is that we derive those ad hoc principles from our formalism. This is an important breakthrough in that we *unify* the scientific theories and show that they are derived from a more fundamental approach. The main challenges of our formalism, as an alternative framework to model dynamics, are figuring out the proper variable, the microstate, to model and the relevant information about the system. Once the relevant information about the system of interest is found, then it is straightforward to manipulate that information and modify the models. In addition, we show that our model derives the stochastic process, which is an advantage of our formalism over the stochastic mechanics where the stochastic process is assumed. Deriving the stochastic process has the advantage of being explicit in what assumptions need to be made to yield such a process.

In this work, we wish to model the dynamics of an exchange rate. Scale invariance is the symmetry of the dynamics of the exchange rate, which should be incorporated in our model. Basically, investors are in favor of investing in securities with higher return than the securities with lower return given that they have the same volatility [40].

It does not matter what the numeric value of the security is, but instead, the return of such a security plays the crucial role. In order to have our formalism be *manifestly* scale invariant, we wish to formulate our formalism such that the probability densities are scalar functions. This choice will lead to choosing the logarithm of the exchange rate as the proper variable to model. Therefore, we want to model how the log exchange rate would change given the current log exchange rate, to be specific, what the transition probability density $P(\ln u^{\prime }|\ln u)$ is, where $u=\frac{S_{f}}{S_{d}}$ represents the current exchange rate of foreign currency to domestic currency and *u*′ is the next exchange rate. It is important to notice that any extension of our model, such as including jumps, should be such that the scale symmetry is not broken, otherwise there will be an arbitrage opportunity.

In Entropic Dynamics, the information about the subject matter takes the form of a constraint equation on the probability density function. Two pieces of information relevant to the dynamics of the log exchange rate are the continuity of the dynamics and the directionality. In this work, we do not take into account that jumps can happen. Including jumps in our model is left for future work. The method of Maximum Entropy, maximizing the relative entropy, is used to assign and update the probability density. By maximizing the relative entropy subject to the constraints, we arrive at the transition probability distribution. The transition probability density will be a Gaussian distribution in the log exchange rate, which amounts to the Geometric Brownian Motion of the exchange rate. Apart from the dynamics of the exchange rate, the dynamics of the probability density is crucial, especially for the purpose of forecasting. We will show that the probability density will evolve according to a Fokker–Planck equation.

In Section 3, we apply our entropic model of the exchange rate to value European Options on the exchange rate from the perspective of a domestic investor. Derivative securities should be valued such that there is no arbitrage opportunity. To establish a no-arbitrage valuation, we derive the risk-neutral probability density. Risk-neutral information is used and imposed on the entropic exchange rate model to derive a risk-neutral measure. The European Options premium is computed by taking the expected values of the Options at maturity and discounting it with a risk free rate. The resulting model is the Garman–Kohlhagen model, which is the counterpart of the Black–Scholes model of European Options on stocks. Then, the call-put parity is derived, which further certifies the no-arbitrage valuation. Using the same procedure, we derive the dynamics of European Options, which is the Black–Scholes–Merton partial differential equation.

## 2. Entropic FX Model

We would like to model how the exchange rate changes. However, as will unfold in the following, the proper variable to model turns out be the logarithm of the exchange rate. Prior information about the subject and the scale invariance symmetry lead us to choose the logarithm of the exchange rate; therefore, we want to model the dynamics of the logarithm of the exchange rate.

Scale invariance is an important symmetry, which ought to be manifest in the dynamics of the exchange rate, namely in the stochastic process that will be derived. Where does this symmetry come from? Investors do no care much about the value of the exchange rate, but if they invest in one, they would like to have a high return from that investment. Therefore, from the investment perspective, the exchange rate with a higher return would be more favorable than the one with a lower return assuming the two assets have the same volatility. If we have two assets, with the same amount of risk associated with them, they are expected to have the same return, otherwise an arbitrage opportunity will emerge. Investors will take advantage of that arbitrage opportunity and equilibrate the market such that the two assets will have the same return. To be more specific, the demand for the asset with higher return will increase, which leads to an increase in the value of that asset. This increase in the value of the asset will lead to a lower return for that asset to the extent that both assets will have the same return. Investors in the market, through the supply and demand forces, will use the arbitrage opportunities to equilibrate the market. This is the essence of the scale invariance.

A simple way of manifesting the scale invariance symmetry is to choose the right function of the exchange rate such that the probability measures are invariant, scalar functions, under the scale transformation. Let us denote the exchange rate as $u=\frac{S_{f}}{S_{d}}$, where $S_{f}$ and $S_{d}$ represent the foreign currency and domestic currency, respectively. The scale transformation is given by,
(1)$$\tilde{u}=l\;u\;$$
where *l* is a positive constant called the scaling factor. We are looking for a function of the exchange rate f(u) such that the probability density $P\left(f(u)\right)$ is invariant under the scale transformation Equation (Equation 1), i.e.,
(2)$$P\left(f(\tilde{u})\right)=P\left(f(u)\right)$$

This leads to a constraint equation for f(u),
(3)$$f\left(\tilde{u}\right)=f\left(u\right)+C$$

Using the scale transformation Equation (Equation 1) twice with two scaling factors *l* and *l*′, we get a constraint on *C*,
(4)$$C\left(l\right)+C\left(l^{\prime }\right)=C\left(l\;l^{\prime }\right)$$

The unique solution to Equation (Equation 4) is C(l)=lnl, which in turn leads to a unique solution to Equation (Equation 3),
(5)f(u)=lnu

Therefore, to have a manifestly scale invariant formalism, we need to choose the logarithm of the exchange rate as our subject matter to model. Notice that once the dynamics of the logarithm of the exchange rate is laid down, we can do any transformation, i.e., any change of variable, to find the dynamics of other functions of the exchange rate.

### 2.1. Statistical Model

The proper variable to model is the logarithm of the exchange rate denoted by $\ln u=\ln\frac{S_{f}}{S_{d}}$. To come up with the dynamics, we address the question: How will the log exchange rate change given the current log exchange rate? In the entropic inference framework, we address such a question by assigning a transition probability distribution, $P(\ln u^{\prime }|\ln u)$. We use the method of maximum entropy to assign the transition distribution,
(6)$$S\left[P,Q\right]=-\int d\ln u^{\prime }\;P\left(\ln u^{\prime }|\ln u\right)\;\ln\frac{P(\ln u^{\prime }|\ln u)}{Q(\ln u^{\prime }|\ln u)}\;,$$
where $Q(\ln u^{\prime }|\ln u)$ is called the prior, which captures prior information when we are making the inference. Next, we specify the prior distribution.

### 2.2. The Prior

The prior distribution can be specified or assigned by using the method of maximum entropy and imposing the prior information. To assign the prior distribution, we maximize the relative entropy S[Q,q] subject to the prior information,
(7)$$S\left[Q,q\right]=-\int d\ln u^{\prime }\;Q\left(\ln u^{\prime }|\ln u\right)\;\ln\frac{Q(\ln u^{\prime }|\ln u)}{q(\ln u^{\prime }|\ln u)}\;,$$
where $q(\ln u^{\prime }|\ln u)$ is a uniform prior, which represents the situation of extreme ignorance. Notice that $q(\ln u^{\prime }|\ln u)$ is a prior for $Q(\ln u^{\prime }|\ln u)$, which itself is a prior for $P(\ln u^{\prime }|\ln u)$. Any non-uniform distribution amounts to prioritizing some outcomes over others, which is in contrast with having no information. The prior information we have about the subject is that the log exchange rate will have a small change, which amounts to saying that the dynamics is continuous. This continuity of dynamics information is represented by the following constraint,
(8)$${\langle {\left(\Delta \ln u\right)}^{2}\rangle }_{Q}={\langle {\left(\ln\frac{u^{\prime }}{u}\right)}^{2}\rangle }_{Q}=k\;,$$
where *k* is small and will be determined in next part. Maximizing the relative entropy Equation (Equation 7) subject to normalization and the continuity constraint Equation (Equation 8), we get the prior distribution,
(9)$$Q\left(\ln u^{\prime }|\ln u\right)=\frac{1}{\eta }\exp\left[-\frac{\alpha }{2}\;{\left(\ln\frac{u^{\prime }}{u}\right)}^{2}\right]$$
where α is large, which will be specified next, and the normalization factor is $\eta =\int_{-\infty }^{\infty }\;d\ln u^{\prime }\;\exp\left[-\frac{\alpha }{2}\;{\left(\ln\frac{u^{\prime }}{u}\right)}^{2}\right]$.

### 2.3. Entropic Time

We need to construct time in our formalism to be able to model the dynamics. Why would we need to construct entropic time? Why can we not use the time that has been used in physics or everyday life? We are putting forth an entropic framework, which is divorced from physics or finance. If our formalism is any good, we ought to be able to model dynamics without resorting to the other theories or formalisms.

The notion of time we introduce here will eventually be the same as the usual conception of time we use to model any stochastic process. Here, we introduce a notion of time that is a convenient tool to keep track of change [11]. We introduce the notion of the *entropic clock*
Δt as following,
(10)$$\alpha \left(\ln u\right)=\frac{1}{\sigma^{2}\left(\ln u\right)\;\Delta t}$$
where $\sigma^{2}$ is the volatility of the log exchange rate. Notice that in Equation (Equation 10), we define α in terms of two variables, which later on are specified as the time duration and volatility. The value for *k* in constraint Equation (Equation 8) can be computed as $k=\frac{1}{\alpha }=\sigma^{2}\;\Delta t$. If volatility were independent of exchange rate, then this notion of time would resemble Newtonian time, otherwise it is similar to a relativistic time. To complete the notion of entropic time, we need to define an *entropic instant*. An entropic instant is defined as,
(11)$$p{\left(\ln u^{\prime }\right)}_{t^{\prime }}=\int d\ln u\;P\left(\ln u^{\prime }|\ln u\right)\;p{\left(\ln u\right)}_{t}\;,$$

If the distribution $p{\left(\ln u\right)}_{t}$ were to represent information at one instant, then the next instance is defined as $p{\left(\ln u^{\prime }\right)}_{t^{\prime }}$, where $t^{\prime }=t+\Delta t$. Notice that an instant is defined by a single state, where in our case, the state is represented by a probability distribution. In addition, Equation (Equation 11) specifies Δt as the time interval; if *t*′ is the next instant to *t*, then the time interval is $t^{\prime }-t=\Delta t$. For simplicity, we write p(lnu,t), instead of $p{\left(\ln u\right)}_{t}$. This parameter *t* has a nice property of being ordered and having an arrow.

### 2.4. The Directionality Constraint

The stochastic process that the prior distribution represents is a Brownian Motion of the log exchange rate with no drift. The new piece of information we have is that there is a drift in the log price. This information is captured in the following constraint,
(12)$${\langle \ln\frac{u^{\prime }}{u}\rangle }_{P}=k^{\prime }\left(u\right)\to 0\;,$$
where $k^{\prime }\left(u\right)$ will be determined shortly. We can Taylor expand the log function,
(13)$$\ln\frac{u^{\prime }}{u}\approx \;\frac{\Delta u}{u}-\frac{1}{2}{\left(\frac{\Delta u}{u}\right)}^{2}\;,$$
and then take the expectation with respect to the posterior distribution,
(14)$${\langle \ln\frac{u^{\prime }}{u}\rangle }_{P}\approx \;{\langle \frac{\Delta u}{u}\rangle }_{P}-\frac{1}{2}{\langle {\left(\frac{\Delta u}{u}\right)}^{2}\rangle }_{P}\;,$$
where the first term of the expansion defines a drift,
(15)$$\langle \frac{\Delta u}{u}\rangle =\left(\mu_{d}-\mu_{f}\right)\;\Delta t\;,$$
where $\mu_{d}-\mu_{f}$ is the difference of domestic and foreign drifts. At this point, we do not need to specify these drifts; however, another model needs to be developed to specify the drifts. To specify the second term of the expansion, we maximize the entropy Equation (Equation 6) subject to normalization and the directionality constraint Equation (Equation 15). This will yield the transition probability,
(16)$$P\left(\ln u^{\prime }|\ln u\right)=\frac{1}{\xi }\;\exp\left[-\frac{\alpha }{2}{\left(\ln\frac{u^{\prime }}{u}\right)}^{2}+\beta \left(u\right)\ln\frac{u^{\prime }}{u}\right]$$
where *β* is a Lagrange multiplier corresponding to the directionality constraint and *α* is given in Equation (Equation 10). The normalization factor is $\xi =\int_{-\infty }^{\infty }\;d\ln u^{\prime }\;\exp\left[-\frac{\alpha }{2}{\left(\ln\frac{u^{\prime }}{u}\right)}^{2}+\beta \left(u\right)\ln\frac{u^{\prime }}{u}\right]$. The transition probability distribution can be rewritten in a Gaussian form,
(17)$$P\left(\ln u^{\prime }|\ln u\right)=\frac{1}{Z(\alpha ,\beta \;,\ln u)}\;\exp\left[-\frac{\alpha }{2}{\left(\ln\frac{u^{\prime }}{u}-\frac{\beta }{\alpha }\right)}^{2}\right]$$
where the new normalization factor is $Z=\int_{-\infty }^{\infty }\;d\ln u^{\prime }\;\exp\left[-\frac{\alpha }{2}{\left(\ln\frac{u^{\prime }}{u}-\frac{\beta }{\alpha }\right)}^{2}\right]$. This transition probability density leads to a Wiener process of the logarithm of the exchange rate,
(18)$$\ln\frac{u^{\prime }}{u}={\langle \ln\frac{u^{\prime }}{u}\rangle }_{P}+\Delta W$$
where the drift and the Brownian Motion are,
(19)$${\langle \ln\frac{u^{\prime }}{u}\rangle }_{P}=\beta \;\sigma^{2}\;\Delta t\;,\;{\langle \Delta W\rangle }_{P}=0\;,\;{\langle {\left(\Delta W\right)}^{2}\rangle }_{P}=\frac{1}{\alpha }=\sigma^{2}\;\Delta t$$

Next, we need to specify the second term in Equation (Equation 14). We take the square of Equation (Equation 13), and taking the expectation, calculation is skipped; we get,
(20)$${\langle {\left(\ln\frac{u^{\prime }}{u}\right)}^{2}\rangle }_{P}={\langle {\left(\frac{\Delta u}{u}\right)}^{2}\rangle }_{P}=\sigma^{2}\;\Delta t$$

Therefore, our directionality constraint is found to be,
(21)$${\langle \ln\frac{u^{\prime }}{u}\rangle }_{P}\approx \;{\langle \frac{\Delta u}{u}\rangle }_{P}-\frac{1}{2}{\langle {\left(\frac{\Delta u}{u}\right)}^{2}\rangle }_{P}=\left(\mu_{d}-\mu_{f}\right)\;\Delta t-\frac{1}{2}\sigma^{2}\;\Delta t=k^{\prime }\;$$

The Lagrange multiplier β can now be specified,
(22)$$\beta =\frac{\left(\mu_{d}-\mu_{f}\right)}{\sigma^{2}}-\frac{1}{2}$$

Summarizing our findings, we get the transition probability density to a be lognormal distribution,
(23)$$P\left(\ln u^{\prime }|\ln u\right)=\frac{1}{Z}\;\exp\left[-\frac{1}{2\;\sigma^{2}\;\Delta t}{\left(\ln\frac{u^{\prime }}{u}-\;\left(\mu_{d}-\mu_{f}-\frac{1}{2}\sigma^{2}\right)\;\Delta t\right)}^{2}\right]$$
with the stochastic process for the log exchange rate as a Brownian Motion with a drift,
(24)$$\ln\frac{u^{\prime }}{u}={\langle \ln\frac{u^{\prime }}{u}\rangle }_{P}+\Delta W$$
(25)$${\langle \ln\frac{u^{\prime }}{u}\rangle }_{P}=\left(\mu_{d}-\mu_{f}-\frac{1}{2}\sigma^{2}\right)\;\Delta t\;,\;{\langle \Delta W\rangle }_{P}=0\;,\;{\langle {\left(\Delta W\right)}^{2}\rangle }_{P}=\sigma^{2}\;\Delta t$$

This is the Brownian Motion for the log exchange rate, which amounts to a Geometric Brownian Motion of the exchange rate. In hindsight, it becomes obvious why we took the expansion in the Taylor expansion only to the second order, simply because the higher orders of expansion are proportional to a higher order of Δt, which in the regime of continuous motion can be neglected.

It is noteworthy to mention that we can observe explicitly that the transition density Equation (Equation 23) is invariant under scaling transformation, i.e., $P\left(\ln u^{\prime }|\ln u\right)=P\left(\ln{\tilde{u}}^{\prime }|\ln\tilde{u}\right)$. Under scaling transformation, the log exchange ratio gets shifted, $\ln\tilde{u}=\ln lu=\ln u+\ln l$, which in turn leads to a shift in the mean of the transition density. Both $\ln u^{\prime }$ and lnu are shifted, and they cancel out, leaving the transition probability density invariant.

### 2.5. Fokker–Planck Equation

Further, we can address how the probability density would evolve over time. Equation (Equation 11) can be written in a differential equation form,
(26)$$\partial_{t}p\left(\ln u,t\right)=-\frac{\partial }{\partial \;\ln u}\left(\left(\mu_{d}-\mu_{f}-\frac{1}{2}\sigma^{2}\right)\;p\left(\ln u,t\right)\right)+\frac{1}{2}\;\frac{\partial^{2}}{\partial {\left(\ln u\right)}^{2}}\left(\sigma^{2}\;p\left(\ln u,t\right)\right)$$

This is the Fokker–Planck equation for the distribution p(lnu,t). If the drifts and the volatility happen to be constant over time and independent of the exchange rate, namely uniform, then for a finite time interval, the transition probability density will be a lognormal distribution of the exchange rate with Δt=T. Notice that the dynamics of the probability density is invariant under the scaling transformation.

## 3. European Options Pricing on FX

European Options are valued in a risk-neutral universe, which is equivalent to a no-arbitrage pricing. In this section, we construct the risk-neural probability distribution and use it to value the European Options on an exchange rate. The model developed is a counterpart of the Black–Scholes model and is known as the German-Kohlhagen model. Next, we derive the Black–Scholes–Merton partial differential equation for the dynamics of European Options.

### 3.1. Garman–Kohlhagen Model

A risk-neutral universe has two main constraints: the expected drift is the same as the risk-free rate, and the rate with which we discount should be the risk-free rate [41]. To drive the risk-neutral measure, we impose the first risk-neutral constraint on Equation (Equation 15),
(27)$${\langle \frac{\Delta u}{u}\rangle }_{P}=\left(r_{d}-r_{f}\right)\;\Delta t\;,$$
where $r_{d}$ and $r_{f}$ are the domestic and foreign risk-free rates. Notice that the derivation of the risk-neutral probability density distribution is the same as the lognormal distribution we derived for the transition probability Equation (Equation 23) with the exception that instead of the drift of the exchange rate, we have the risk-free rates. Further, we assume that the risk-free rate and the volatility are uniform, and we get the risk-neutral measure,
(28)$$P\left(\ln u^{\prime }|\ln u\right)=\frac{1}{Z}\;\exp\left[-\frac{1}{2\;\sigma^{2}\;\Delta t}{\left(\ln\frac{u^{\prime }}{u}-\;\left(r_{d}-r_{f}-\frac{1}{2}\sigma^{2}\right)\;\Delta t\right)}^{2}\right]$$

Notice that with the assumption of the uniformity of the risk-free rate and the volatility, this is the risk-neutral transition probability for any finite time interval Δt. If we relax these assumptions, we need to solve the Fokker–Planck equation to solve for the risk-neutral distribution at any time in the future.

Now, we can proceed to value European Options with the risk-neutral measure. We simply compute the expected payoff of the Options at maturity and discount it using a risk-free rate to get the premium. The expected payoff of a call Option, denoted by $V_{c}$, for a domestic investor at maturity is given by the difference between the expected sale value and the expected purchase value,
(29)$$V_{c}={\langle \mathrm{Sale}\rangle }_{LN,T}-{\langle \mathrm{Purchase}\rangle }_{LN,T}$$
where we have,
(30)$$\begin{array}{cc} {\langle \mathrm{Sale}\rangle }_{T}= & \int_{K}^{\infty }\;du\;P\left(u,T|u_{0}\right)\;u\\ {\langle \mathrm{Purchase}\rangle }_{T}= & \int_{K}^{\infty }\;du\;P\left(u,T|u_{0}\right)\;K \end{array}$$
where *K* is the strike exchange rate and $u_{0}$ is the current exchange rate. We integrate from the strike rate since, if the rate is less than the strike rate, we will not exercise the call Option. Then, the payoff can be written as,
(31)$$V_{c}=\int_{K}^{\infty }\;du\;P\left(u,T|u_{0}\right)\;\left(u-K\right)$$

The Premium for the call Option is just the discounted value of the payoff, we discount the expected payoff with the domestic risk-free interest rate,
(32)$$C=e^{-r_{d}T}\;V_{c}=e^{-r_{d}T}\;{\langle \mathrm{Sale}\rangle }_{LN,T}-e^{-r_{d}T}{\langle \mathrm{Purchase}\rangle }_{LN,T}$$

Since we know the current exchange rate $u_{0}$, then the risk-neutral probability distribution at maturity is given by,
(33)$$\begin{array}{cc} P(u_{T}|u_{0}) & =\int \;d\tilde{u}\;P\left(u_{T}|\tilde{u}\right)\;P\left(\tilde{u}|u_{0}\right)=\int \;d\tilde{u}\;P\left(u_{T}|\tilde{u}\right)\;\delta \left(\tilde{u}-u_{0}\right)\\ & ∼LN(\ln u_{0}+\left(r_{d}-r_{f}\right)\;T-\frac{\sigma^{2}\;T}{2}\;,\;\sigma \;\sqrt{T}) \end{array}$$
where LN stands for lognormal distribution. Next, we compute the expected sale value at maturity, and we get,
(34)$${\langle \mathrm{Sale}\rangle }_{LN,T}=u_{0}\;\exp\left[\left(r_{d}-r_{f}\right)\;T\right]\;N\left(d_{1}\right)$$
where $d_{1}=\frac{\ln u_{0}+\left(r_{d}-r_{f}\right)\;T-\frac{\sigma^{2}T}{2}-\ln K}{\sigma \;\sqrt{T}}$ and $N(d_{1})$ is the standard normal cumulative distribution function,
(35)$$N\left(d_{1}\right)=\frac{1}{\sqrt{2\pi }}\int_{-\infty }^{d_{1}}dx\;e^{-\frac{x^{2}}{2}}$$

The expected purchase value can be computed,
(36)$${\langle \mathrm{Purchase}\rangle }_{LN,T}=KN\left(d_{2}\right)$$
where $d_{2}+\sigma \;\sqrt{T}=d_{1}$. Then, the premium of the call Option is:(37)$$C=u_{0}\;e^{-r_{f}T}N\left(d_{1}\right)-e^{-r_{d}T}\;K\;N\left(d_{2}\right)$$

This is the celebrated Garman–Kohlhagen model for the call Option. To value a European put Option, we follow the same procedure as the call Option. The expected payoff at the maturity for the put Option is,
(38)$$V_{p}=\int_{\infty }^{K}\;du\;P\left(u,T|u_{0}\right)\;\left(u-K\right)$$

Notice that we integrate to the strike rate *K* because if the rate is greater than the strike rate, we will not exercise the put Option. Discounting this expected payoff will yield the put premium,
(39)$$P=e^{-r_{d}T}\;V_{p}=e^{-r_{d}T}\;K\;N\left(-d_{2}\right)-u_{0}\;e^{-r_{f}T}\;N\left(-d_{1}\right)$$

Which is the Garman–Kohlhagen model for the European put Option. We can simply check that the call and put premium satisfy the so-called call-put parity relation,
(40)$$C-P=e^{-r_{f}T}\;u_{0}-e^{-r_{d}T}\;K$$

The call-put parity relation ensures that this was a no-arbitrage valuation of European Options.

### 3.2. BSM Differential Equation

We can drive the differential equation for the European Options, which is known as the Black–Scholes–Merton differential equation. To derive the differential equation, we start with the expected payoff equation,
(41)$$V\left(\ln u,K,t\right)=\int d\ln u_{T}\;P\left(\ln u_{T},T|\ln u,t\right)\;\left(u_{T}-K\right)\;$$

The boundaries of the integral are left out to get the differential equation for both call and put Options. Next, we take the time derivative of both sides,
(42)$$\partial_{t}V=\int d\ln u_{T}\;\left(u_{T}-K\right)\;\partial_{t}P\left(\ln u_{T},T|\ln u,t\right)\;$$
where the time derivative of the transition probability is given by the backward Kolmogorov equation; the derivation is skipped,
(43)$$\partial_{t}P\left(\ln u_{T},T|\ln u,t\right)=-\left(\left(r_{d}-r_{f}\right)-\frac{\sigma^{2}}{2}\right)\frac{\partial P(\ln u_{T},T|\ln u,t)}{\partial \ln u}-\frac{\sigma^{2}}{2}\frac{\partial^{2}P\left(\ln u_{T},T|\ln u,t\right)}{\partial {\left(\ln u\right)}^{2}}\;$$

Substituting Equation (Equation 43) into Equation (Equation 42), we get,
(44)$$\begin{array}{cc} \partial_{t}V= & \int d\ln u_{T}\;\left(u_{T}-K\right)\left[-\left(\left(r_{d}-r_{f}\right)-\frac{\sigma^{2}}{2}\right)\frac{\partial P}{\partial \ln u}-\frac{\sigma^{2}}{2}\frac{\partial^{2}P}{\partial {\left(\ln u\right)}^{2}}\right]\\ = & -\left(\left(r_{d}-r_{f}\right)-\frac{\sigma^{2}}{2}\right)\;\frac{\partial }{\partial \ln u}\;\int d\ln u_{T}\;\left(u_{T}-K\right)\;P\left(\ln u_{T},T|\ln u,t\right)\\ & -\frac{\sigma^{2}}{2}\;\frac{\partial^{2}}{\partial {\left(\ln u\right)}^{2}}\;\int d\ln u_{T}\;\left(u_{T}-K\right)\;P\left(\ln u_{T},T|\ln u,t\right)\\ = & -\left(\left(r_{d}-r_{f}\right)-\frac{\sigma^{2}}{2}\right)\;\frac{\partial V}{\partial \ln u}-\frac{\sigma^{2}}{2}\;\frac{\partial^{2}V}{\partial {\left(\ln u\right)}^{2}} \end{array}$$

We can rewrite this equation as,
(45)$$\partial_{t}V\left(u,t\right)+\left(r_{d}-r_{f}\right)\;u\;\frac{\partial V}{\partial u}+\frac{\sigma^{2}\;u^{2}}{2}\;\frac{\partial^{2}V}{\partial u^{2}}=0$$

The partial differential equation for the European Option premium is derived just by substituting $E=e^{r_{d}\left(t-T\right)}\;V$ into the above equation,
(46)$$\partial_{t}E+\left(r_{d}-r_{f}\right)\;u\;\frac{\partial E}{\partial u}+\frac{1}{2}\;\sigma^{2}\;u^{2}\;\frac{\partial^{2}E}{\partial u^{2}}-r_{d}\;E=0$$
where by applying the boundary conditions for the call/put Option, we get the solution for the call/put Option.

## 4. Summary and Discussion

We put forth an entropic framework to model the dynamics of exchange rates. This alternative framework is complementary to the stochastic process modeling because we derived the stochastic dynamics from maximizing a relative entropy. To derive the transition distribution, we needed to take into account the relevant information. The scale symmetry of the dynamics is a significant piece of information, which led us to choose the logarithm of the exchange rate as the proper variable to model. The continuity of the motion and the directionality were the other pieces of information we had about the exchange rates, which were formulated as a constraint equation. The resulting model was a Geometric Brownian Motion for the exchange rate. Further, a dynamics for the probability density distribution was found to be a Fokker–Planck equation.

Next, we applied the entropic exchange rate model to value European Options on FX. We derived the risk-neutral probability density by imposing the risk-neutral constraints. Using the risk-neutral measure, we valued the European Options, which was the same as the known Garman–Kohlhagen model. The dynamics of the European Options was found to be the Black–Scholes–Merton partial differential equation.

An extension to our model could be allowing the dynamics to have jumps. We imposed the continuity of the motion to derive the Geometric Brownian Motion of the exchange rate. By taking into account the information about the jump process, we can extend our model, which further will lead to a modified Options value.

We have extended our framework to model the dynamics of stocks and valuing European Options on stocks [40]. It was shown that under similar constraints, we could yield a Geometric Brownian Motion of stocks. Then, a no-arbitrage pricing of European Options on the stock was provided, which gave rise to the Black–Scholes model, and the dynamics of the Option premium was found to be the Black–Scholes–Merton differential equation. Another extension could be modeling the dynamics of many stocks [10].

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
