# Peer review of "Entropic Dynamics of Exchange Rates and Options"

_entropy, 2019, doi:10.3390/e21060586_

Round 1

Reviewer 1 Report

The authors derive Fokker-Plank equation for the exchange rate between foreign and domestic currencies. They use the so-called entropic inference to achieve this goal.

I have a background in physics, and I found the paper interesting as an introduction to the statistical analysis of exchange rates. As far as I can see, the most important point of this analysis is that one should treat the logarithm of the exchange rate as the relevant parameter of the theory, but not the exchange rate itself.

Concerning the mathematical aspects of the paper, I do not see much novelty in it. The authors emphasize entropic inference and similar concepts. Ultimately, they derive the Fokker-Planck equation (26), which is known to be universal and valid for any stochastic process. However, there is no need to re-derive Eq. (26) since it has been already derived in a very general form, and applies to any processes with small jumps in the relevant parameter during a single time step.

In conclusion, the paper may be published as a nice review and introduction to the statistics of finance for non-experts. However, it is not novel enough to be counted as an original research paper.

Author Response

Please find attached the pdf file.

Reviewer 2 Report

The authors use the relative entropy and its maximization to re-derive the Garman-Kohlhagen model of call and put options of European currencies. They also derive the Black-Scholes-Merton equation for the payoff.

The paper is fairly well-written although some points, detailed below, were not clear to me. I should say that I am not an expert in econophysics, so some of my questions may simply reflect my ignorance. These are my questions:

1) In Eq. (2) what is P? The authors later use "P" to denote a transition probability and "p" to denote a probability density. Is "P" in Eq. (2) a probability density?

2) Eq. (3) implies that P(f) = P(f+C). What is the meaning of this equation? Why does it imply scale invariance?

3) In Eq. (8) what is "k"? Is it just a constant?

4) As far as I could see, Eq. (10) does not clearly define "Delta t". Does Eq. (10) define "alpha" or does it define "Delta t"? Is "Delta t" simply the time period between the purchase of the option and its execution?

5) In Equation (11) why is "p" named "entropic instant" instead of simply "probability density". What is the use of calling it "entropic instant"? Is t' = t + Delta t?

6) In Eq. (12) the quantity k'(u) approaches zero. When does it approach zero? Does it approach zero as Delta t approaches zero? The authors say that they will determine k'(u) "shortly" but they never do so.

7) In Eq. (17) what is "Z"?

8) In Eq, (33) does "LN" mean "LogNormal"?

9) The idea of deriving probability distributions from maximization of entropy is well-known. The derivation of the Garman-Kohlhagen and Black-Scholes-Merton equation are also well-know. Can the authors elaborate on what is the novelty of their approach, as compared with other approaches? What would be an advantage of their approach to derive new results, as compared with other approaches? For example, what would be the advantage of their approach when modeling jumps or multiple stocks?

I ask the authors to incorporate answers to these questions in the paper,  assuming the questions are not trivial.

Author Response

Please find attached the pdf file.

Round 2

Reviewer 2 Report

The authors answered my questions so now I recommend the paper for publication. There are two additional points I noticed:

1) On lines 154, 184 and 186, the integration variable is missing

2) In their reply letter, the authors included the equation

P (ln u' | ln u) = P (ln v' | ln v)

where "v" is "u tilde". This equation helped me understand better what the authors were trying to say, regarding the scale invariance. I suggest the authors include this equation after the sentence on lines 199, 200.

Author Response

Hello,

Please find attached the pdf file.

Best
